# Legal Aspect of Plastic Waste Management in Indonesia and Malaysia: Addressing Marine Plastic Debris

**Hanim Kamaruddin [1,\*], Maskun [2], Farida Patittingi [3], Hasbi Assidiq [4], Siti Nurhaliza Bachril [4] and Nurul Habaib Al Mukarramah [4]**

1   Faculty of Law, Universiti Kebangsaan Malaysia, Bangi 43600, Malaysia
2   International Law Department, Faculty of Law, Hasanuddin University, Kota Makassar 90245, Indonesia; maskunmaskun31@gmail.com
3   Civil Law Department, Faculty of Law, Hasanuddin University, Kota Makassar 90245, Indonesia; farida.pada@unhas.ac.id
4   Faculty of Law, Hasanuddin University, Kota Makassar 90245, Indonesia; assidiqhasbi97@gmail.com (H.A.); nurhalizabachr@gmail.com (S.N.B.); habaib.edu@gmail.com (N.H.A.M.)
\*   Correspondence: hanim@ukm.edu.my

**Abstract:** Marine plastic debris is a common issue faced by the entire international community, with some countries finding it exceptionally difficult to address and combat the issue, including Indonesia and Malaysia. The two neighboring countries are ranked as the largest contributors of plastic waste in the ocean. Unmanaged plastic waste that will ultimately end up in waters and seas has become a major issue that Indonesia and Malaysia must deal with, and a firm legal approach holds a key role for both countries in solving the issue. This paper is devoted to normatively analyzing the various legal approaches that are/can be used by Indonesia and Malaysia, and to identify the problems related to such approaches. This article applies normative legal research, in which various legal instruments and other secondary legal materials are descriptively analyzed to unravel the legal issues. The main findings reveal that laws and regulations, as well as public policies that serve as a legal basis and approaches to deal with plastic waste governance in Indonesia and Malaysia, still possess some weaknesses, including the absence of distinctive provisions specifically aimed at dealing with plastic waste, the lower legal binding power of some instruments due to their soft-law nature, and the application of rather ineffective approaches. One important initial step towards actually exerting the legal approaches in governing plastic waste in both countries is to strengthen the governing structure and legal culture surrounding the management of plastic waste. Finally, this paper encourages the establishment of a bilateral agreement between Indonesia and Malaysia that will allow both countries to formulate a more legally binding framework for tackling the issues of marine waste in general and marine plastic debris in particular.

**Keywords:** marine plastic debris; Indonesia; Malaysia; legal and policy approaches

## 1. Introduction

The use of plastic in people's daily lives has been steadily increasing in an ever-widening range of applications. Plastic continues to be perceived as a more accessible, more durable, lighter in weight, and cheaper material for various purposes, and therefore is often the preferred option for many [1,2]. Since the plastics industry began, total global plastic production has increased exponentially from 1.5 million metric tons in 1950 to more than 9 billion metric tons in 2017 [3]. Global plastic production is expected to keep escalating [4], with product packaging dominating the primary use of plastic [5].

Unfortunately, the rapid increase in plastic production and the wide use of plastic products have not been followed by proper management of their disposal. Most plastic waste that exists in the environment is not well managed or even managed at all. This situation is common in many countries, with Indonesia and Malaysia being no exception.

A study has revealed that the two neighboring countries are among those that contribute the most plastic debris that ends up in the ocean. In fact, 6 out of the 11 Southeast Asian member states are on the list [6].

A report by the Indonesia National Plastic Action Partnership (NPAP) revealed that around 4.8 million tons (70%) of all plastic waste in Indonesia are not managed. It is estimated that 0.62 million tons (9%) of all unmanaged plastic waste ultimately end up in Indonesian waters and seas [7]. Related data highlighted by the Indonesian Institute of Sciences (LIPI) state that around 0.27–0.60 million tons of plastic waste are leaked into Indonesian seas every year [8]. Similar conditions also occur in Malaysia, where it is estimated that only 8.4% of all generated plastic waste is recycled, while 75.8% is dumped in landfills and open spaces [9]. One study suggested that Malaysia has generated 0.94 million tons of unmanaged plastic waste, 0.14 to 0.37 million tons of which may have been washed into the ocean [6].

Indonesia and Malaysia have shared marine areas and water bodies (as shown in Figure 1), and thus, the transboundary movement of marine debris between the two countries is inevitable. Furthermore, uncontrollable stranded debris originating from elsewhere, including plastic imports, intensifies the challenges faced by Indonesia and Malaysia, placing more pressure on their domestic plastic debris management. A study by Cordova et al. (2020) pointed out the lack of sufficient legal and policy mechanisms as one of the reasons driving the unsolved problems of transboundary marine debris [10].

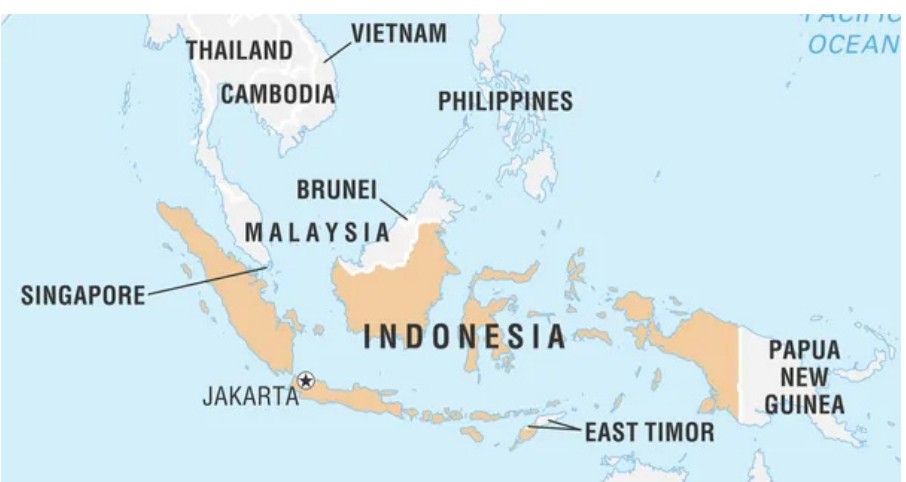

**Figure 1.** Simple map of Southeast Asia. Source: https://www.britannica.com/place/Indonesia/Celebes-and-the-Moluccas (accessed on 20 May 2022).

By putting forward marine plastic debris as a common issue faced by Indonesia and Malaysia, this paper intends to analyze the legal approaches that are/can be used by both countries in addressing and dealing with the issue, and further, to identify the problems related to such approaches. It is also devoted to providing a scientific basis for policy making related to the management of plastic debris in the marine environment, both on a national scale, particularly for Indonesia and Malaysia, and on a regional scale, especially for the Association of Southeast Asian Nations (ASEAN). Furthermore, this research is expected to be able to improve on prior comparable studies related to legal and policy approaches for managing plastic debris in the ocean.

In the first part, this paper will describe the sources and impacts of plastic waste in Indonesia and Malaysia. The second part will explicate the legal approaches that both countries possess in addressing and combating the plastic waste issue, including the approaches derived from instruments under the ASEAN approach, since both countries are members of the organization. Lastly, the third part will serve an analysis of the legal issues revolving around the efforts of combating the plastic waste that both Indonesia and

Malaysia are currently facing, and how strengthening the legal system can be a way to resolve the issue.

## 2. Materials and Methods

This research applied normative-legal research method carried out by utilizing a statutory approach that seeks to review relevant materials on plastic waste in the marine areas surrounding Malaysia and Indonesia. The data used are primary legal materials obtained from relevant laws and regulations as well as public policies, and secondary legal materials obtained from various related literatures on both countries. These are analyzed and assessed to reveal the issues and recommend solutions concerning plastic waste management in Indonesia and Malaysia.

The analysis and assessment began by conducting a legislative inventory in organizing primary legal materials (legal instruments) that had a direct, indirect, or probable relationship with management of marine plastic debris within both Indonesian and Malaysian legal framework. Since both countries are parties to the Association of Southeast Asian Nations (ASEAN), the legislative inventory also included some relevant instruments established by the ASEAN. The primary legal materials that were used and analyzed in this paper are listed in Table 1. They were organized and analyzed according to their year of entry into force, and the object they cover from the general to the more specific object. For Indonesia in particular, hierarchical order of laws and regulations is put as the main consideration in organizing and analyzing the materials.

**Table 1.** Primary legal materials.

| | Instrument | Year | Source of Retrieval |
|---|---|---|---|
| Regional Level | ASEAN Framework of Action on Marine Debris | 2019 | The ASEAN Secretariat: https://asean.org/asean2020/wp-content/uploads/2021/01/3.-ASEAN-Framework-of-Action-on-Marine-Debris-FINAL.pdf (accessed on 4 January 2021). |
| | Bangkok Declaration on Combating Marine Debris in ASEAN Region | 2019 | The ASEAN Secretariat: https://asean.org/asean2020/wp-content/uploads/2021/01/2.-Bangkok-Declaration-on-Combating-Marine-Debris-in-ASEAN-Region-FINAL-1.pdf (accessed on 4 January 2021). |
| | Regional Action Plan for Combating Marine Debris in the ASEAN Member States 2021–2025 | 2021 | The ASEAN Secretariat: https://asean.org/book/asean-regional-action-plan-for-combating-marine-debris-in-the-asean-member-states-2021-2025-2/ (accessed on 23 May 2022). |
| | Indonesia | | |
| National Level | Law Number 18 Year 2008 on Waste Management | 2008 | Legal Documentation and Information Network (LDIN/JDIH), the House of Representatives of the Republic of Indonesia: https://www.dpr.go.id/jdih/index/id/145 (accessed on 4 February 2020). |
| | Government Regulation Number 81 Year 2012 on Household Waste and Household like Waste Management | 2012 | LDIN/JDIH, the Audit Board of the Republic of Indonesia: https://peraturan.bpk.go.id/Home/Details/5295/pp-no-81-tahun-2012 (accessed on 4 February 2020). |
| | Government Regulation Number 27 Year 2020 on Specific Waste Management | 2020 | LDIN/JDIH, Ministry of Environment and Forestry of the Republic of Indonesia: http://jdih.menlhk.co.id/uploads/files/PP_Nomor_27_Tahun_2020_menlhk_06222020120956.pdf (accessed on 19 December 2020). |
| | Presidential Regulation Number 97 Year 2017 on National Policy and Strategy for Household Waste and Household-like Waste Management | 2017 | Information System, Ministry of Law and Human Rights of the Republic of Indonesia: http://peraturan.go.id/common/dokumen/ln/2017/ps97-2017.pdf (accessed on 10 January 2021). |
| | Presidential Regulation Number 83 Year 2018 on Marine Waste Management | 2018 | LDIN/JDIH, Ministry of Marine and Fishery Affairs of the Republic of Indonesia: https://kkp.go.id/djprl/jaskel/artikel/20399-peraturan-presiden-republik-indonesia-nomor-83-tahun-2018-tentang-penanganan-sampah-laut (accessed on 10 January 2021). |
| | Ministry of Environment and Forestry Regulation Number 75 Year 2019 on the Roadmap of Waste Reduction by Producers | 2019 | LDIN/JDIH, Ministry of Environment and Forestry of the Republic of Indonesia: http://jdih.menlhk.co.id/uploads/files/P_75_2019_PETA_JALAN_SAMPAH_menlhk_12162019142914.pdf (accessed on 8 July 2020). |
| | Law Number 23 Year 2014 on Regional Government (amended) | 2014 | LDIN/JDIH, the Audit Board of the Republic of Indonesia: https://peraturan.bpk.go.id/Home/Details/38685/uu-no-23-tahun-2014 (accessed on 17 October 2021). |
| | Law Number 32 Year 2009 on Environmental Protection and Management (amended) | 2009 | LDIN/JDIH, the Audit Board of the Republic of Indonesia: https://peraturan.bpk.go.id/Home/Details/38771/uu-no-32-tahun-2009 (accessed on 17 October 2021). |

**Table 1.** *Cont.*

| Instrument | Year | Source of Retrieval |
|---|---|---|
| | | Malaysia |
| Laws of Malaysia Act 127 (Environmental Quality Act 1974) (amended) | 1974 | Ministry of Energy and Natural Resources of Malaysia: https://wk.ketsa.gov.my/sites/NREDrive/UndangUndang/UNDANGUNDANG/Akta%20Kualiti%20Alam%20Sekeliling%201974%20[Akta%20127].pdf (accessed on 16 February 2022). |
| Laws of Malaysia Act 671 (Solid Waste and Public Cleansing Management Act 2007) | 2007 | National Solid Waste Management Department, Ministry of Housing and Local Government of Malaysia: https://jpspn.kpkt.gov.my/resources/index/user_1/Perundangan/Akta-akta/act672bi.pdf (accessed on 16 February 2022). |
| Malaysia's Roadmap towards Zero Single Use Plastics 2018–2030 | 2018 | Ministry of Education of Malaysia: https://www.moe.gov.my/images/KPM/UKK/2019/06_Jun/Malaysia-Roadmap-Towards-Zero-Single-Use-Plastics-2018-2030.pdf (accessed on 16 February 2022). |
| National Marine Litter Policy and Action Plan 2021–2030 | | Ministry of Environment and Water of Malaysia: https://www.kasa.gov.my/resources/alam-sekitar/national-marine-litter-policy/31/ (accessed on 23 May 2022). |

In order to provide an overview of the current condition of waste management and related policies implementation in both Indonesia and Malaysia, a study on different literatures as the secondary legal materials were carried out. The literature mainly includes journal articles, briefs published by both governmental and non-governmental institutions, and news articles. In addition, statistical records related to the number of waste generation and unmanaged waste from both countries were also incorporated in order to present more concrete descriptions.

## 3. Results

### 3.1. Plastic Debris' Sources and Impacts: Indonesia and Malaysia

3.1.1. Sources and Leakage of Plastic Debris to the Ocean

It has been observed that both Malaysia and Indonesia have similar diversification of plastic waste sources. In this case, Malaysia's plastic production has more varied segmentation whereas Indonesia provides for Fast-Moving Consumer Goods (FMCG) and Consumer Goods (CG) to fulfill plastic production demand from consumers. This correlates to the rising rate of household consumption expenditure in Indonesia (6.072 quadrillion according to The World Bank) [11]. Such production has contributed to plastic waste addition, which has been discovered in 2019 that 46% of discarded plastic waste are dominated by several FMCG and CG brands such as Unilever and Indofood [12]. The comparison of plastic waste sources and number of plastic waste between Indonesia and Malaysia are shown in Table 2.

**Table 2.** Comparison of Malaysia and Indonesia in sources of plastic waste, mismanaged plastic waste and its leakage to waterways and ocean.

| Aspects | Indonesia | Malaysia |
|---|---|---|
| Sources of plastic waste | Industrial (plastic production) sector [13] | Plastic production (purposed for agriculture, household, packaging, construction, electronics, automotive, and other subsector) [14] |
| | household waste [15] | household waste [14] |
| | municipal solid waste [15] | municipal solid waste [14] |
| | plastic waste importation [16] | plastic waste importation [14] |
| Number of plastic waste production | 7.8 million tons/year [7] | 1.27 million tons/year [6] |
| Number of mismanaged plastic waste | 4.8 million tons (70% of total plastic waste) [7] | 0.94 million tons (74% of total plastic waste) [6] |
| Numbers of plastic waste leakage to waterways and ocean | 0.62 million tons (9% of total mismanaged plastic waste) [7] | 0.37 million tons (40% of total mismanaged plastic waste) [6] |

The consumer sector, which consists of household waste and municipal solid waste also become the source of plastic waste for both countries. In Indonesia, each person in household sector contributes 0.22–0.4 kg waste generation every day, meanwhile municipal solid waste (including traditional market, commercial area, public facilities, and other areas) contributes 3,4 million metric tons of plastic waste generation [15]. Similar case

occur in Malaysia, where household sector waste generation tends to contribute higher than in Indonesia by 0.85–1.5 kg/person/day where 74% of the waste constituted plastic waste [14].

Geographical basis between Malaysia and Indonesia as a neighboring country which simultaneously put both countries in the global south has become the ultimate reason on how global south countries tend to be targeted as plastic waste destinations by global north countries such as the United States of America, Australia, and the United Kingdom [17]. The routes of plastic waste exportation to Indonesia and Malaysia can be seen in Figure 2.

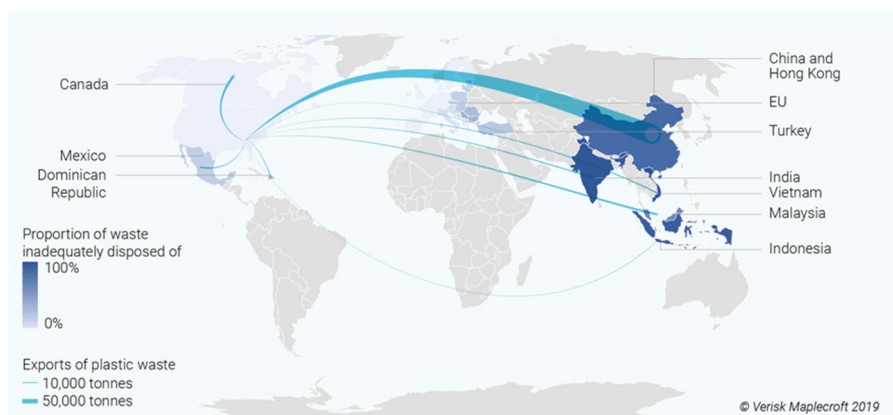

**Figure 2.** Plastic waste exportation map. Source: https://www.maplecroft.com/insights/analysis/where-should-the-worlds-waste-go/ (accessed on 22 May 2021).

By observing the imported plastic waste issue, Indonesia reaches 474,649.628 kg on plastic waste importation which become an additional task for Indonesia to manage such waste apart from the national plastic pollution issue [16]. This issue occurs due to companies and industries within Indonesia tend to choose cheaper option that relies to countries plastic waste importation rather than buying directly from local scavenger which allows high flow of plastic waste importation to Indonesia [18].

Similarly, Malaysia also has become transitory and destination country of waste importation. Despite the number of waste importations (in total of 105.000 tons in 2017) tends to be lower than plastic waste import, Malaysia also faced the issue of illegal plastic waste import containing low grade unrecyclable plastics by labelling carriages with codes that does not need any permit where it would end up burned illegally and releasing toxic substances [14]. One particular example of illegal waste import in Indonesia is the entry of 584 carriages containing waste that are mixed with plastic waste and hazardous waste, according to a waste import supervision report by the Director General of Customs and Excise, where 450 of the carriage have been re-exported [19]. Nevertheless, the remaining illegal waste import still able to cause hazard to environment by untreated improper waste management that could lead to leakage onto water and polluting the marine.

### 3.1.2. Challenges and Impact

After analyzing the sources of plastic waste in both countries, there are several underlying challenges both countries face in tackling plastic debris issue which become an obstruction to sustainable development:

1    High rate of consumption, especially in household and municipal sector;
2    Inadequate waste management issue which could led to significant amount of mismanaged waste leaked into waterways and ocean;
3    Obstruction of both countries' struggle in tackling national plastic pollution issue are due to being targeted as destination for plastic waste import.

Such aforementioned challenges which led plastic waste ended up becoming plastic debris in the ocean, begin to bring harmful impact to Indonesia's continuity of marine biodiversity, and seafood consumption safety leading into high economic cost a nation

should paid for both countries. Several unique marine animals in Indonesia currently contaminated with plastic, such as how an ancient fish *Coelacanth* found out dead due to plastic contamination in its digestion tract, numerous whales died and drifted away to the shore with identified plastic debris contamination in its organs, and other 267 marine species exposed to danger due to plastic waste consumption [20]. The seafood safe-consumption aspect is being impacted due to several regions such as Bali that starts to found increasing prevalence of microplastics in four different types of fish which has a high and pivotal economic value in Bali seafood market [21]. Tourism attraction also highly impacted by mismanaged plastic waste, where 70% out of 30 tons plastic waste are collected on the shore in one of Indonesia's top tourists destinations, Kuta Beach [22]. By myriads of negative impacts occurs due to plastic debris in marine environment, it can be observed that this issue impacts not only polluters, but also the whole economic and environment continuity.

Within 4675 km of coastline containing rich marine biodiversity, Malaysia's abundance of biodiversity is endangered due to the presence of this marine debris, specifically plastic marine debris. The risk on decreasing tourism attraction from recreational beach become even clear, which proved by a study that in 2021 there are 36.895 kg of waste collected from 1463 km of Malaysian coast which consists dominantly by food wrappers and plastic items including but not limited to bottles, caps, lids, bags, and cups that accumulates into 304.656 items [23].

Furthermore, the plastic waste drifted away to ocean and become marine debris would be fragmented into microplastics and bring dangerous exposure to marine life and seafoods that often consumed by local society. The first identification of shellfish toxins occurs in Malaysian water, where this toxin is produced by floating plastic debris that then contaminated with harmful pathogens that would bring hazardous impacts if consumed by marine organism and human [24]. Similar cases also happen in consumed seafoods such as anchovy that have been detected with microplastics contained in 80% of its liver in Malaysian markets, where this proves the aforementioned argument on how plastic debris bring further danger to human [25]. All the impacts of plastic debris, also entails to high economic cost where it is estimated that APEC region including Malaysia would experience $10.8 billion of marine economy damage cost from marine debris [26].

*3.2. Legal Approaches on Marine Plastic Debris Management by ASEAN, Indonesia, and Malaysia*

3.2.1. ASEAN Legal Approach

In addressing marine debris-related issue within the region of ASEAN, an attention should be devoted to the two current and most specified ASEAN legal instrument addressing the issue, which are the ASEAN Framework of Action on Marine Debris (ASEAN Framework) and the Bangkok Declaration on Combating Marine Debris in ASEAN Region (Bangkok Declaration). In addition, approaches and measures proposed in Regional Action Plan for Combating Marine Debris in the ASEAN Member States 2021–2025 (RAP 2021–2025) are also worth to become consideration.

(1)    ASEAN Framework [27]—*A Guide for Concrete Regional Actions*

The establishment of ASEAN framework stems from the awareness of marine debris being a transboundary issue which requires integrated regional cooperation to be resolved. ASEAN Member States (AMS) putting the issue in urgency, see the strong collaborations that are particularly crucial to tackle marine debris problem within the region, and envisioning: "*without immediate action, marine debris pollution may negatively impact marine biodiversity, environment, health, society and economy*". The Framework encompasses four priority areas, where each comprises different actions and suggested activities envisaged to combat marine debris pollution. Further elaboration on the priority areas along with actions and suggested activities is provided in Table 3.

**Table 3.** Legal Approaches on Marine Plastic Debris Management by ASEAN, Indonesia, and Malaysia.

| Instrument | Approaches |
|---|---|
| **ASEAN Legal Approaches** | |
| ASEAN Framework | → Encompasses four priority areas: (i) Policy Support and Planning; (ii) Research, Innovation, and Capacity Building; (iii) Public Awareness, Education, and Outreach; and (iv) Private Sector Engagement,<br>→ Formulates 13 actions and about 44 suggested activities including to (i) develop and implement ex-tended producer responsibility (EPR), (ii) encourage national authorities to collaborate with businesses in developing and promoting the product sustainability and circularity criteria while concurrently addressing the unsustainable use and disposal of single-use plastic products, develop/strengthen upstream policies for land-based and sea-based leakage including single-use plastics, and (iii) enhance research/study on plastics and microplastics,<br>→ Reinforces the implementation of relevant international laws and agreements, one of which is UN Environment Assembly resolutions 3/7 on Marine Litter and Microplastics. |
| Bangkok Declaration | → Recalls the implementation of the United Nations 2030 Agenda for Sustainable Development, in particular to comply with Goal 14: Conserve and sustainably use the oceans, seas and marine resources for sustainable development, which through its Target 14.1 call for actions to "prevent and significantly reduce marine pollution of all kinds . . . " including by partnering with stakeholders at relevant levels to address their production, marketing and use of plastics and microplastics,<br>→ Reaffirms prior commitments especially ones stated in the EAS Conference on Combating Marine Plastic Debris and the recommendations from the ASEAN Conference on Reducing Marine Debris in ASEAN Region; and the East Asia Summit Leaders' Statement on Combating Marine Plastic Debris which seeks to take concrete actions in combating marine plastic debris, in particular to strengthen regional and international cooperation,<br>→ Promotes innovative solutions in enhancing plastics value chains and improving resource efficiency by prioritizing approaches such as circular economy and 3R (reduce, reuse, recycle),<br>→ Welcomes the effort for capacity building and exchange of best practices among the ASEAN member states as well as support from external partners in this regard. |
| Regional Action Plan 2021–2025 | → Sets the same priority areas as ASEAN Framework<br>→ Designates 3 elements of the waste value chain: (1) Reducing inputs into the system; (2) Enhancing collection and minimizing leakage; (3) Creating value for waste reuse,<br>→ Proposes 14 regional actions, including the establishment of: (i) guiding principles for phasing out select single-use plastics, (ii) regional guidebook on standards for responsible plastic waste trade, sorted plastic waste and recycled plastics, (iii) a regional platform to support innovation and investments in plastics and plastic waste management, and so on. |
| **Indonesian Legal Approaches** | |
| Law 18/2008 | Establishes the general framework for waste management in Indonesia. Waste management is conducted using 2 mechanisms, which are 'reducing' and 'handling'. The waste reduction mechanism consists of 3 activities, which are: (a) limitation of waste generation; (b) waste recycling; and/or (c) reuse of waste. Meanwhile, the handling mechanism comprises 5 actions, they are: (i) waste sorting; (ii) waste collecting; (iii) waste transporting; (iv) waste processing; and/or (v) waste final processing. |
| GR 81/2012 | |
| GR 27/2020 | |
| PR 97/2017 | Sets out a 30% target or about 20.9 million tons of waste reduction in Indonesia by 2025 and a 70% target or 49.9 million tons of waste handling by 2025. |
| MEF Regulation 75/2019 | → Follows the same framework for waste reduction mechanism established in the Law 18/2008, the GR 81/2012, and the GR 27/2020 which consist of: (a) limitation of waste generation, (b) waste recycling, and (c) reuse of waste,<br>→ Obligates producers to collect the waste originated from their products, products packaging, and/or containers as an additional procedure following the recycle and reuse mechanism. In doing so, producers are then required to provide the collection facilities,<br>→ According to the Roadmap (attached in Annex I), waste reduction by producers is arranged in 5 stages of action:Planning, consist of drafting of planning documents, development of facility and mechanism as well as collaboration, and trial of reduction actions;<br>→ Implementation, comprising the execution of waste reduction based on planning document;<br>→ Supervision;<br>→ Evaluation; andReportAll these actions will be conducted within a 10-year period, from 2020 to 2029. |
| PR 83/2018 (NAP on Marine Waste Management) | → Puts forward 5 strategies: (i) National movement to raise awareness among stakeholders; (ii) Land-based waste management; (iii) Handling of waste in coastal and marine areas; iv) Funding mechanisms, institutional reinforcement, supervision, and law enforcement; and v) Research and development,<br>→ Strategies are broken down into 13 programs and about 57 activities with specific objectives that will be conducted in different locations. Each of the said activities possesses some ambitious outputs to achieve and designated periods of time for various stages of implementation,<br>→ The NAP is set up to perform within 8 years, from 2018 to 2025,A large part of the activities focuses on downstream approaches which in general encompass the control and handling of plastic waste, most of the approaches are intended to intervene the land-based plastic waste sources,<br>→ Encourages the establishment of international cooperation in dealing with the issue, it even specifies the output to be an international agreement regarding the handling of marine plastic debris. |
| Regional-level Regulations | Impose either restrictions or total ban on the use of single-use plastics, especially plastic bags. |
| **Malaysian Legal Approaches** | |
| EQA 1974 | → Establishes the general norm stipulating restrictions that no person shall, unless licensed, discharge environmentally hazardous substances, pollutants or wastes into any lands, inland waters, the territorial waters of Malaysian; violation to such restrictions will lead to sanctions of fine or imprisonment. |

**Table 3.** *Cont.*

| Instrument | | Approaches |
|---|---|---|
| SWMA 2007 | → | Sets a licensing mechanism for every person who wishes to: (1) Undertake or provide any solid waste management services; (2) Manage or operate any solid waste management facilities; or (3) Undertake or provide any public cleansing management services, |
| | → | Provides the Director General of Solid Waste and Public Cleansing Management with the responsibility for solid waste and public cleansing management- (i) to set standards, specifications and codes of practice relating to any aspect of solid waste management services and public cleansing management services; (ii) giving approvals for construction of any solid waste management facilities and (iii) require the use of any method or manner for the purpose of reducing the adverse impact of the controlled solid waste on the environment. |
| Roadmap towards Zero Single Use Plastics 2018–2030 | → | Comprises a set of action plans that will be conducted in 3 phases. Phase 1 (2018–2021) provides planned action ranging from individual-based approaches such as practicing 'no straw by default' and imposing pollution charge, to policy intervention such as reviewing existing laws or developing legal framework on single-use plastics. Phase 2 (2022–2025) and Phase 3 (2026–2030) consist of continuation of activities from Phase 1, |
| | → | Considerably important action includes incorporating plastic waste governance in a binding legal instrument through the introduction of a legal framework on single-use plastics which will take place within Phase 2. |
| NMLPAP 2021–2030 | → | Designates five priority pillars: (1) Policy adoption and implementation; (2) Deployment of technologies, innovation and capacity building; (3) Improve monitoring and data collection on marine litter; (4) CEPA (communication, education & public awareness) and outreach; and (5) Adopt whole-of-nation and multi-stakeholders' approach in harmonizing cross-cutting objectives, |
| | → | Sets out six desired national outcomes, one of which is deployment of solutions including latest technology and innovation to tackle marine plastic pollution, through a phased approach, |
| | → | Specifies 17 identified national actions and 103 key activities to be implemented in a phased approach: short-term (2021–2023), medium-term (2024–2027), and long-term (2028–2030). |

(2) Bangkok Declaration [28]—*Reinforcing the Commitments*

Bangkok Declaration is a reaffirmation and reinforcement of previous commitments to which the AMS have been bound. The Declaration emphasizes the ASEAN Community Vision 2025, in particular, the ASEAN Socio-Cultural Community (ASCC) Blueprint 2025 on Conservation and Sustainable Management of Biodiversity and Natural Resources. It is indeed also a reaffirmation to the commitment of strategic measures to "*promote cooperation for the protection, restoration and sustainable use of coastal and marine environment, respond and deal with the risk of pollution and threats to marine ecosystem and coastal environment, in particular in respect of ecologically sensitive areas*".

The primary concern that Bangkok Declaration tries to put forward in justifying the efforts of combating marine plastic debris, as stipulated by the Declaration itself, is the high and rapidly increasing levels of marine debris in particular marine plastic litter and the expected increase in negative effect on mostly marine-related entities. An attention shall also be devoted to the urgent need for strengthened knowledge of the levels and effects of microplastics and nano-plastics on marine ecosystem, food safety and human health. Highlights of approaches created in Bangkok Declaration are provided in Table 3.

(3) Regional Action Plan for Combating Marine Debris in the ASEAN Member States 2021–2025 [29]

The RAP 2021–2025, as it claimed, aims to improve coordination at the regional and international levels in achieving a sustainable management of coastal and marine environment by addressing marine plastic pollution. The RAP's objectives are aligned with the eight objectives under the Bangkok Declaration, with a specific focus on marine plastic debris, which will be operationalized across the four components outlined in the ASEAN Framework.

The RAP outlines a series of actions designed to tackle the plastic waste littering and marine debris issues in the ASEAN, it is also commits to bring the vision of a more sustainable approach to plastics to fruition. The identified measures are intended to make significant progress during the plan's validity period while also laying the groundwork for longer-term action. Highlights of approaches set in the RAP can be found in Table 3.

3.2.2. Indonesian Legal Approaches for Waste Management

(1)   General Legal Framework for Waste Management

Laws and regulations dealing with waste management, ranging from national to regional level have been exist in Indonesia. However, none of these legal instruments are aimed to specifically govern plastic waste and/or plastic waste pollution. In specific, the term 'plastic waste' is not recognized under the provisions of these laws and regulations. Therefore, 'plastic waste' can be identified by the character of "difficult to be decomposed under natural processes" which appears throughout articles and paragraphs of the instruments, or by the terms "reusable waste" and "recyclable waste".

The primary legal foundation for waste management in Indonesia is the Law of the Republic of Indonesia Number 18 Year 2008 on Waste Management (Law 18/2008, or the Waste Management Law). To further elaborate and implement the provisions of the Law, several derivative regulations were then established, including Government Regulation of the Republic of Indonesia Number 81 Year 2012 on Household Waste and Household-like Waste Management (GR 81/2012) and Government Regulation of the Republic of Indonesia Number 27 Year 2020 on Specific Waste Management (GR 27/2020). While the legal instruments may have different specific provisions due to the distinguished type of waste they deal with, they all, in principle, have a uniformed framework which derives from rules established in the Law 18/2008 [30–32].

In later development, all the efforts of reducing and managing the waste as comprised in the laws and regulations are then led to achieve targets put forward in National Policy and Strategy for Household Waste and Household-like Waste Management (known as Jakstranas in Indonesian) which is enacted through Presidential Regulation of the Republic of Indonesia Number 97 Year 2017 (PR 97/2017). Jakstranas is Indonesia's primary plan for waste management which will be implemented within the period of 2017–2025 [33]. Detailed information on legal framework established by the Law 18/2008, the GR 81/201, and the GR 27/2020 as well as targets set forward by the Jakstranas are shown in Table 3.

(2)   Waste Reduction by Producers—*Embracing the Business Actors*

The involvement of producers in waste management circle is pertinent to ensure a progressive step towards the efforts of embracing all sectors to extend and intensify the works of waste management in Indonesia. Their roles are regulated through Ministry of Environment and Forestry of the Republic of Indonesia Regulation Number 75 Year 2019 on the Roadmap of Waste Reduction by Producers (MEF Regulation 75/2019).

The MEF Regulation 75/2019 adopts the spirit of Extended Producers Responsibility (EPR) Principle [34]. Under the Regulation, 'producers' are defined as "business actors who produce, distribute (including goods originated from imports), or sell goods with packaging or containers that cannot or are difficult to decompose by natural processes." The reducing mechanism in MEF Regulation 75/2019 is specifically devised to deal with products, products packaging, and/or containers that are: (a) hard to decompose under natural process; (b) un-recyclable; and/or (c) un-reusable, including those made from plastic materials [35]. The heart of MEF Regulation 75/2019 lies in its Annex, specifically Annex I, which is the roadmap itself. Further explanation regarding the roadmap is served in Table 3.

(3)   Regional-level Regulations

At the regional level governments in Indonesia, province and regency/municipality, are mandated with several tasks and functions over waste management. Such powers and functions are delegated by related instruments including the Law 18/2008, the Law on Regional Government (Law Number 23 Year 2014 (revised)), and the Law on Environmental Protection and Management (Law Number 32 Year 2009 (revised)).

Amongst powers held by regional-level governments, it is purposed to establish policies and strategies [30]. In relation to their powers to establish policies and strategies over waste management within their area of administration, a number of regulations

have been established by regional-level governments that aimed specifically to deal with circulation and generation of plastic and/or plastic waste within their region.

According to the Ministry of Environment and Forestry of the Republic of Indonesia as reported by Al Faqir (2021), as of January 2021, there are 2 provinces and 39 regencies/municipalities, or in total 41 regions in Indonesia that have already enacted regulations and/or policies imposing restriction or ban on the use of single-use plastics [36]. Examples of the regional-level regulations in question are Bali Governor Regulation Number 97 of 2018 on Restrictions on the Generation of Single-use Plastic Waste (provincial-level), and Balikpapan Mayor Regulation Number 8 of 2018 on Reduction of Plastic Bags Usage (municipality-level).

(4)    National Action Plan on Marine Waste Management [37]—*What it has to Offer*

Presidential Regulation of the Republic of Indonesia Number 83 Year 2018 on Marine Waste Management (PR 83/2018) is a progressive national legislation that is specifically enacted for marine waste management and is segregated from governance of other kinds of waste in Indonesia. The scope is intelligible and distinctive. With the regulation enacted, it shows that the Indonesian Government had at least taken a major first step in addressing marine waste issue: set it down in a binding legal instrument. One of the primary considerations of putting the management of marine debris in a distinctive regulation by enacting the PR 83/2018 is to comply with the Government commitment to handling plastic waste debris up to 70% in 2025.

The heart of PR 83/2018 lies in its inseparable part- the National Action Plan (NAP) on Marine Waste Management, an arrangement document that serves as a guidance for relevant ministry or other government' non-ministerial institution as well as a reference for the community in taking steps towards the acceleration of marine waste control. Some noteworthy activities listed in the NAP include establishment of a unit for utilization of plastic waste as fuel oil, production of prototype equipment for waste power plant (WPP), utilization of plastic waste as additive for road construction in the form of plastic asphalt [38]. Detailed elaboration on PR 83/2018 along with its NAP are shown in Table 3.

3.2.3. Malaysian Legal Approach

Like Indonesia, Malaysia also does not possess a uniform approach in addressing plastic waste and plastic pollution, including and particularly ones resulting from consumption of single-use plastics. Neither does it have any specific laws and regulations to govern such matters [39]. The Environmental Quality Act 1974 (EQA) for instance, as the main legislation in preventing or controlling pollution in Malaysia, does not specifically recognize the term 'plastic waste' under its provisions. However, this sort of waste could be surmised into the Acts' definition of waste in general. It defines: *'waste includes as any matter prescribed to be scheduled waste, or any matter whether it is in a solid, semi-solid or liquid form, or in the form of gas or vapor which is emitted, discharged or deposited in the environment . . . '* [40].

Another primary legal basis dealing with waste management in Malaysia is Solid Waste and Public Cleansing Management Act 2007 (Act 672/SWMA 2007). Categories of waste governed by this Act are divided into: commercial solid waste, construction solid waste, public solid waste, household solid waste, industrial solid waste, imported solid waste, institutional solid waste, special solid waste, controlled solid waste, and recyclable solid waste [41].

Apart from the SWMA 2007, there are no other binding legal instruments that aim to govern solid waste in Malaysia, let alone plastic waste including pollution resulting from solid waste. However, several environmentally sound waste management policies that could be used as basis and guide for waste management in Malaysia including the 1998 Action Plan for a Beautiful and Clean Malaysia (ABC), the 2005 National Strategic Plan for Solid Waste Management (NSP), the 2006 Master Plan on National Waste Minimization (MWM), the 2006 National Solid Waste Management Policy, the Solid Waste Corporation Strategic Plan (2009–2013) and the Tenth Malaysian Plan (2011–2015) that strengthen and

further develop Malaysia's sustainable waste management practices [42]. Highlights of legal approaches established by the EQA 1974 and the SWMA 2007 can be found in Table 3.

(i)     Towards Zero Single-Use Plastics—*Stepping Up from Status Quo*

Despite the absence of a binding legal instrument addressing plastic waste and pollution issues, Malaysia is currently undertaking a significant move by implementing The Roadmap towards Zero Single Use Plastics 2018–2030 which serves as a guide for stakeholders in taking concrete actions towards eradication of single-use plastics. The Roadmap was launched by the then Ministry of Energy, Science, Technology, Environment & Climate Change (MESTECC) of Malaysia in 2018. It is expected to provide a policy direction with a more holistic manner and approach in combating the issue of plastic waste in lieu of inexistent uniformity in legal approach. The Roadmap is to be implemented from 2018 to 2030 with an expectation that all relevant stakeholders will play their roles effectively to ensure the objectives of the Roadmap are met [43]. Further elaboration regarding the Roadmap can be found in Table 3.

(ii)     National Marine Litter Policy and Action Plan 2021–2030

National Marine Litter Policy and Action Plan (NMLPAP) 2021–2030 serves as a broad-based implementation plan to address marine litter pollution (including plastic pollution) guided by a multi-stakeholder approach. It is intended to coordinate actions to address marine litter pollution at the national, state and local levels in accordance with international standards and approaches and, as appropriate, in harmony with programmes and measures applied in the region. The NMLPAP 2021–2030 highlights priority areas and actions to address marine litter pollution in Malaysia. It will be implemented alongside the National Roadmap Towards Zero Single-Use Plastics 2018–2030 [44]. Highlights of the NMLPAP 2021–2030 are provided in Table 3.

## 4. Discussion

### *4.1. Weaknesses and Challenges of Current Legal Approaches*

#### 4.1.1. ASEAN Context

Weaknesses of approaches and challenges for implementation revolving around The ASEAN Framework of Action on Marine Debris (ASEAN Framework) and the Bangkok Declaration on Combating Marine Debris in ASEAN Region (Bangkok Declaration) as the main legal instruments for regional actions dealing with marine debris will primarily be resulted from the soft law nature of these instruments. As soft laws are characterized by lesser degrees of precision, obligation and delegation of extensive powers [45], and commonly associated with an attenuated or non-existent binding force [46], the ability of the ASEAN-generated instruments in general to deal with regional issues and situations has repeatedly been called into question due to those comprehensions [47].

A study by Mahaseth and Subramaniam (2021) revealed that the lack of clarity stems from the prevalence of 'soft law' nomenclature in ASEAN-generated instruments (agreements) has resulted into the ASEAN members states (AMS)' lack of call for 'hard law' obligations. As an implication, while the various ASEAN instruments have been framed, their implementation is hampered [47]. The ASEAN Framework and the Bangkok Declaration, along with the Regional Action Plan for Combating Marine Debris in the ASEAN Member States 2021–2025 are, by their nature, not legally binding. Therefore, they do not give rise to legal obligations for the AMS. Kadarudin et al. (2020) suggested that an upgrade to the two regional instruments in order to transform them into hard law would be of necessity in the future, it will allow them to be legally binding for the AMS' by ratification [48].

#### 4.1.2. Indonesian and Malaysian Contexts

As mentioned earlier, laws and regulations as well as public policies related to the management of waste established within national legal framework of Indonesia and Malaysia do not recognize the term 'plastic waste' in their provisions, nor do they segregate the

governance of plastic waste from other kinds of waste, and thus, do not apply any differentiated framework in treating the waste of the sort. Accordingly, the governance of plastic waste will fall under the established framework explained above, as with all types of waste in general. In other words, there is no specific treatment or distinctive management system for plastic waste in both Indonesia and Malaysia. While Indonesia and Malaysia share initial challenges in terms of the recognition and distinctive management system of plastic waste and let alone marine plastic debris within their national legislations, a major different of legal approaches dealing with waste management from both countries can be first spotted in the number of instruments they have, where Indonesia has significantly more laws and regulations concerning waste management.

Problems of having less legally binding instruments guiding the management of plastic debris become more challenging for Malaysia since the Solid Waste and Public Cleansing Management Act (SWMA) 2007, despite being the only significant legislation available to regulate waste, unfortunately does not contain provisions directed to minimize or prevent plastic waste and/or plastic waste pollution. Rather, its provisions only cover administrative mechanisms of waste management in Malaysia. Part X of SWMA 2007 indeed contains provisions on reduction and recovery of controlled solid waste. Section 101 in this part deals with reduction, reuse and recycling of controlled solid waste [41]. Nevertheless, in substance, the article merely contains provisions giving power to the Director General of Solid Waste and Public Cleansing Management to give order or to set requirements for such reduction, reuse, and recycling actions to be implemented by related stakeholders. None of the provisions set a technical guide or any sort of guide on how to conduct reduction, reuse, and recycling efforts.

Provisions regarding 'take back system and deposit refund system' are contained in Section 102 of SWMA 2007, solely give powers to the Minister to, by order, establish deposit refund system and determine: (a) the specified products or goods; (b) the deposit refund amount; (c) the labelling of the products or goods; and (d) the obligations of the dealers of the products or goods. It does not necessarily regulate the mechanism of 'take back and deposit refund system' [41]. Thus, questions have been drawn as to whether the available laws and regulations are sufficient to effectively address the issues of plastic waste especially marine plastic debris in this case. A study by Van et al. (2021) for instance, revealed that existing legislations are perceived as not significantly affect single-use plastic reduction behavior in Malaysia. Majority of respondents involved in the study disagreed that existing laws and regulations are able to decrease usage of single-use plastics [49].

Other than problems related to the availability of legal instruments, challenges hindering efforts to reduce and control plastic waste generation will likely be found in the implementation of Malaysia's Roadmap towards Zero Single Use Plastics 2018–2030 which will undoubtedly require huge public participation. A study by Chen et al. (2021) revealed that the single-use plastic ban policy is moderately received by the public. It suggested that people may need time to adjust with such a restriction since plastic is a common product for most people due to its beneficial features. This study, for instance, showed how plastic straw ban in Kuala Lumpur, Putrajaya, and Selangor back in 2019 received a lukewarm response from citizens. Government's will to reduce single-use plastic usage will also be hampered due to a relatively low recycling rate which is resulting from recycling cost that is higher than the cost of purchasing newly manufactured plastics [14].

In addition to challenges related to implementation, when it comes to Roadmap towards Zero Single Use Plastics 2018–2030 and the NMLPAP 2021–2030, the fact that they are categorized as soft laws which will have implications to lessening their legal binding power cannot be ruled out. Overall, the absence of a uniform approach to deal with waste -especially plastic waste- may lead to different implementations and performances in parts of country's regions [50]. As of 2019, only 6 states (Perlis, Kedah, Melaka, Negeri Sembilan, Pahang and Johor, as well as 2 federal territories, Putrajaya and Labuan) have enforced the Act the SWMA 2007 [14] that will subsequently lead to legislation including action plans and strategies to be overall less effective to reduce plastic litter [50].



Indonesia, despite having quantitatively more legal instruments that serve as basis justifying actions to deal with marine plastic debris, is still nowhere near ideal in addressing the issue. Some significant weaknesses of current legal approaches could be identified within the Presidential Regulation of the Republic of Indonesia Number 83 Year 2018 (PR 83/2018) on Marine Waste Management along with its National Actions Plans (NAP) as a rather sophisticated legal instrument aimed in specific to deal with marine debris problems.

Vast majority of the activities comprised in the NAP, approximately 41 activities, are aimed specifically to deal with plastic waste issues, ranging from educational activities to policy intervention efforts. Certain activities listed in the NAP are already well known in Indonesian basic rules for waste management where some resemble the old yet less efficient approaches such as the reduce-reuse-recycle (3R) method and coast clean-up actions. However, this sort of activity is still lacking in quality, as seen in the NAP arrangement, but is also in terms of scale and frequency. The national action for coasts and small islands clean-up for example, will only be conducted in 160 nationwide events within the 8 years period or in equal only 20 events per-year. Similar program listed namely the national movement for coasts and sea clean-up, will only be conducted in 24 locations all over Indonesia. One major question worth asking: will such activities be sufficient and work effectively with adequate funding?

### 4.2. Strengthening the Governing Structure and Legal Culture

Other than robust legal approaches, an ideal governing structure also plays pivotal roles in bringing appropriate and effective plastic waste management into reality, this mainly points out towards policy makers and enforcers. In the ASEAN scope, a number of working groups are already established to lead the implementations of ASEAN initiatives related to plastic waste management, including the ASEAN Working Group on Coastal and Marine Environment (AWGCME) as the lead working group that coordinates issues related to marine debris, the ASEAN Working Group on Environmentally Sustainable Cities (AWGESC), the ASEAN Working Group on Chemical and Waste (AWGCW), and the ASEAN Working Group on Environmental Education (AWGEE) [29].

In national scope, Indonesia for instance, a national coordinating team for the handling of marine debris, chaired by the Coordinating Minister for Maritime Affairs was established to implement the designated activities listed in the NAP on Marine Waste Management. More than 16 ministers and other non-ministerial government agency leaders are positioned in the team structure [37]. This shows how structural composition of the organs mandated to solve the marine debris problems in Indonesia, let alone the waste issue in general, is quite complex. Similar structure could also be found in the regional-level related institutions.

Despite the existence of the required bodies in charge of enforcing all the designated policies dealing with marine plastic debris, it will take a lot of effort and political will of these enforcers to actually realize the objectives and targets of such regulations and policies. An open monitoring scheme to identify and track the work progress of each regulation and policy enforcers, either by individual or institutional, could be of consideration in ensuring the effective implementation. Without any viable monitoring mechanism, it will be difficult to assess whether responsible agencies or stakeholders have implemented the regulations and policies, as well as the action plans and programs. Therefore, an appropriate monitoring measures or mechanisms will help strengthening and safeguarding the legal structure of marine debris management.

In addition to the ideal governing structure, a significant transformation in legal culture also becomes a highly important element in addressing plastic waste issue. In the context of legal culture, community behavior towards the legal regime and its system shall be considered, including beliefs, values, ideas and hopes for the law itself where legal behavior is closely related to the awareness towards emerging issues. In ASEAN region, the reluctant behavior towards eliminating the use of plastic products is due to habitual

nature of consumers and dependency on plastic usage. This is also worsened by the low recycling rate for plastic waste in most countries in ASEAN regions [29].

In Indonesia, the legal culture related to management of plastic waste is not too different from other member states of ASEAN. Policies related to the prohibition of the use of single-use plastics are often opposed by business and industrial actors, and even by the government official who find the restrictions hindering the growth of plastics industry. The drafting of policies related to public participation in reducing plastic litter is lacking [51], where this is crucial in increasing the potential for the success of any existing policies to reduce plastic litter.

Malaysia has similar concerns in the legal culture towards plastic waste management. For instance, recycling rate in Malaysia is considerably low which is only at 28% [50]. This may reflect the existing legislation that do not provide for sufficient recycling mechanism directives to ensure that all stakeholders reduce use of plastic in addressing plastic litter in the ecosystem. Several solutions have been proposed from time to time, such as presenting alternatives to plastic and managing plastic waste based on a circular economy approach [14]. Thus, it is important to realize the compliance to existing laws and regulations as well as public policies relating to plastic waste management within the community. Awareness and consciousness towards the plastic consumption and their disposal post-consumption, as well as the detrimental impact of the unmanaged plastic disposals should also be a priority target by the laws and policy makers.

*4.3. Probable Bilateral Agreement*

In the pursuance of a more robust legal approach, this paper would like to recommend the adoption of a bilateral agreement between Indonesia and Malaysia in which joint efforts from both countries in managing the marine debris in general and marine plastic debris in particular could be negotiated and formulated. The establishment of such an agreement, which means incorporating the current and/or later-developed approaches into a 'hard law' instrument, can be considered a milestone for stepping up from the current regional legal approaches that are still derived from less binding legal instruments as explained in Section 4.1. A study by Duvic-Paoli (2021) revealed that soft law scholars have argued that when used in conjunction with an international agreement, a non-binding norm may lose its non-binding character [52].

Legal justifications for the bilateral agreement in question can be found in a number of legal instruments explained earlier in Section 3.2. Bangkok Declaration, for instance, encourages the enhancement of regional and international cooperation including on relevant policy dialogue and information sharing [28]. In a more binding obligation scheme, Indonesia through the PR 83/2018 and its NAP on Marine Waste Management indeed urges the arrangement on regulation in supporting the completion of the NAP in form of international agreements, which further is specified to be 'An international agreement on the prevention of plastic waste in the sea which includes transboundary issues' [37].

## 5. Conclusions

A number of existing legal instruments could be used by Indonesia and Malaysia in dealing with the issues of plastic waste in general, and marine plastic debris in particular. In the regional level, several ASEAN-generated instruments have specifically created as a guidance on marine debris management within the Southeast Asian Nations, namely the ASEAN Framework of Action on Marine Debris and the Bangkok Declaration on Combating Marine Debris in ASEAN Region, along with the Regional Action Plan for Combating Marine Debris in the ASEAN Member States 2021–2025. At national level, both Indonesia and Malaysia have a set of legal instruments in regulating the management of their domestic waste. The legal instruments in questions include laws and regulations, as well as some national roadmap and action plans.

Despite of the varied legislations, it is apparent that the current legal approaches still possess some weaknesses which will ultimately challenge their implementation. The

weaknesses in question include: (a) Less legal binding power of some instruments due to their soft-law nature, especially embodied in the ASEAN-generated instruments; (b) The absence of uniformed approach and distinctive provisions specifically aimed to regulate the management of plastic waste, where both Indonesia and Malaysia do not yet have a specific legislation which puts plastic waste as its main object of regulation. Malaysia, in particular, still struggle with the unavailable uniformed approach in addressing plastic waste issue. The adoption of two national actions plans for the handling of single-use plastics and marine litter is deemed progressive. Nevertheless, the implementation of the two action plans would likely to face challenges due to the perception and legal culture of community towards the use of plastic materials. There is also possibility that the plan implementation will be hampered because of the 'soft law' nature; and (c) The resemblance of rather ineffective approaches, that are mostly found within Indonesian legislations especially the PR 83/2018 and its NAP on Marine Waste Management.

Besides the adoption of specific legal approaches, strengthening governing structure and legal culture surrounding the management of plastic waste become another important element in order to achieve an effective management for marine plastic debris. Finally, by putting forward marine plastic debris as a common issue faced by Indonesia and Malaysia, and considering proximity of the two neighboring countries especially their geographical conditions -e.g., bordering land and sea areas, this paper strongly encourages the adoption of a bilateral agreement between Indonesia and Malaysia. The establishment of such agreement will allow the management and regulation of marine debris by a legally binding instruments, which subsequently will create legal obligations for both countries.

**Author Contributions:** Conceptualization, H.K. and M.; methodology, H.A.; software, S.N.B.; validation, M. and N.H.A.M.; formal analysis, N.H.A.M.; investigation, H.A.; resources, S.N.B.; data curation, M.; writing—original draft preparation, N.H.A.M., H.K. and S.N.B.; writing—review and editing, M., S.N.B. and N.H.A.M.; visualization, M.; supervision, M. and F.P.; project administration, M., H.K. and S.N.B.; funding acquisition, H.K. All authors have read and agreed to the published version of the manuscript.

**Funding:** This research is financially supported by research funds from Universiti Kebangsaan Malaysia (UKM) Research Grant Fund (GUP-2019-064).

**Institutional Review Board Statement:** Not applicable.

**Conflicts of Interest:** The authors declare no conflict of interest. The funders had no role in the design of the study; in the collection, analyses, or interpretation of data; in the writing of the manuscript, or in the decision to publish the results.

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
