# Peer review of "Legal Aspect of Plastic Waste Management in Indonesia and Malaysia: Addressing Marine Plastic Debris"

_sustainability, doi:10.3390/su14126985_

Round 1
Reviewer 1 Report
This is a sound article. It is well researched and well written. I believe more could and should be done in explaining the implications of the legal approaches and findings. The authors use "legal" a lot (legal culture, legal structures etc), but seem to stop in there. In particular, the Conclusion should be longer, with more implications on what the findings from this study mean for the field. Secondly, the study requires better embeddedness in the broader context. How do Malaysia and Indonesia compare with other countries in the region? How do efforts of ASEAN compare with other regional efforts, e.g. MERCOSUR, NAFTA and - most interestingly the EU which is presumably most advanced in the regulation of plastic? Some engagement with that is needed and literature is available. Finally, the abstract shoudl be imporoved. State the "some weaknesses" which are mentioned. What does it mean to strengthen legal system and to establish a legally binding cooperation between the two neighboring countries? How would that look like and what would it improve?
Reviewer 2 Report
The manuscript ID "sustainability-1709346" focuses on legal aspects of plastic waste management in Indonesia and Malaysia. It is a well structured paper and clearly depict the legal aspect of the plastic waste management.
My suggestion is to improve the quality of the manuscript regarding the following features:
- in keywords avoid the use of terms already used in the title
- english language,
- bibliography formatting (e.g. line 34)
- in material and methods please report year ranges of literature search as well as legal documents
- subsections numbering of results and discussion
- wide the conclusion section
Reviewer 3 Report
This is a thoughtful and generally well-informed paper on a topic of growing importance. The fact that it focuses on efforts to address the issue of marine plastic debris in SE Asia is a distinct plus.
I feel this paper has the makings of a publishable article. At the same time, I think there is considerable room for improvement of the current draft. Addressing the issues I list below would make for a much stronger published article.
- The paper emphasizes legal measures, which is understandable given that the authors are associated with law faculties. Nevertheless, it seems important to recognize the relevance of a variety of policy instruments ranging from hard law to soft law and on to measures that aren't really legal in the ordinary sense of the term. From the text, I infer that a diverse collection of measures are relevant to the challenge of addressing marine plastic debris in the relevant region. But the paper lumps them together under the rubric of legal measures. It would help to provide a more nuanced treatment.
- My understanding is that ASEAN generally proceeds through the use of fairly informal measures. What is the actual status of the Bangkok Declaration in this regard? This does not mean that such measures are inappropriate or ineffective. But they do differ from standard legal measures. To me, this raises serious questions about the idea of a Convention on Marine Plastic Debris in ASEAN (pg. 11).
- Then, there are questions about implementation or moving legal and other measures from paper to practice. The paper has relatively little to say about the challenges of implementing policy instruments in the relevant issue area. But this is a critical issue in terms of likely outcomes.
- The paper speaks at length about policy measures in Indonesia and Malaysia. This is certainly appropriate. But I'm wondering whether there are opportunities for bilateral initiatives in this realm between the two countries, especially since they share closely connected marine areas.
- In this regard, it would help to include a map allowing the reader to visualize the relationship between the land areas of the two countries and the marine areas between them.
- The English in the paper is good enough so that there is no difficulty in following the argument. But there are lots of little English-language problems. Thus, the paper needs copyediting by someone who can fix these problems.
Reviewer 4 Report
The authors presented a manuscript entitled "Legal Aspect of Plastic Waste Management in Indonesia and Malaysia: Addressing Marine Plastic Debris" aiming at analyzing various legal approaches that are and/or can be used by Indonesia and Malaysia, and to identify problems related to such approaches.
The results and conclusion of the manuscript do not prove the acomplished of the main objective.
The attached file highlight the necessary improvements for the entire work, from abstract to conclusions in order to make it worth of publication.

Round 2
Reviewer 3 Report
This is a revised version of a paper I reviewed in its original form.
As before the topic is important and deserves to be raised and discussed widely. So, I have no doubt about the suitability of a paper on this theme to be published in Sustainability.
I have read the authors' responses to my comments and looked at the revised manuscript to see how their revisions have addressed my concerns. I am happy to say that the new version seems substantially improved. The authors have made a serious effort to strengthen their analysis, particularly when it comes to the strengths and weaknesses of various types of soft law instruments and to the issues associated with moving governance arrangements from paper to practice.
I am ready to recommend the acceptance of this version of the paper for publication.
Reviewer 4 Report
The authors have made significant improvements on the manuscript.